# Differences in Tumor Growth and Differentiation in NSG and Humanized-BLT Mice; Analysis of Human vs. Humanized-BLT-Derived NK Expansion and Functions

**DOI:** 10.3390/cancers15010112

**Published:** 2022-12-24

**Authors:** Kawaljit Kaur, Anahid Jewett

**Affiliations:** 1Division of Oral Biology and Oral Medicine, University of California, School of Dentistry and Medicine, Los Angeles, CA 90095, USA; 2The Jane and Jerry Weintraub Center for Reconstructive Biotechnology, University of California, School of Dentistry and Medicine, Los Angeles, CA 90095, USA; 3The Jonsson Comprehensive Cancer Center, University of California, Los Angeles, CA 90095, USA

**Keywords:** IFN-γ, NK cells, cytotoxicity, cancer, humanized-BLT (hu-BLT) mice, NSG mice, osteoclasts, OC-expanded super-charged NK cells

## Abstract

**Simple Summary:**

Cancer biologists have used NSG mice for many years as preclinical model to study the biology and the success of therapeutic modalities for cancer. Although these mice are easy to use for the implantation of tumors, they do not fully represent the human disease, since they do not have the key immune effectors that control the growth and expansion of tumors. In contrast, humanized-BLT mice have these key immune effectors, and they resemble humans in their immune cells, with some differences. We compared the growth, expansion, and infiltration of immune cells in tumors and their function in tumors implanted in these two types of mice. We also studied the immune cells from humanized-BLT mice and compared these to human immune cells. Our studies show that humanized-BLT mice is an appropriate model to study human cancer, especially for studies of Natural Killer cells.

**Abstract:**

There is significant interest and debate regarding the best mouse model of human disease, since studies in wild-type mice may not always recapitulate human diseases. The NSG mouse model has been one of the most commonly used mouse models to study cancer; however, this mouse model, even though it has several advantages in regard to the ease of tumor implantation and financial feasibility, does not represent human disease due to the immunodeficient nature of this model. In this study, we performed oral and pancreatic tumor studies in NSG and hu-BLT mice and found several distinguishing features that make hu-BLT model more suitable for studying human cancer. In addition, we compared the immune function of humans to hu-BLT mice to understand the differences and similarities of the models. Oral and pancreatic cancer stem cells were implanted in NSG and hu-BLT mice. Both tumors grew robustly in NSG mice and killed them within a short period of time. On the contrary, unlike NSG mice, tumor-bearing hu-BLT mice survived longer, grew smaller tumors, and the grown tumors exhibited lower rates of expansion, with a higher surface expression of MHC-class I and lower NK cell-mediated cytotoxicity that was previously shown to have more of a differentiated phenotype. Although the peripheral blood of hu-BLT mice in comparison to that of humans had lower percentages of NK cells and cytotoxic function, it mediated a higher secretion of IFN-γ, likely contributing to the differentiation of the tumor cells and subsequent decrease in the tumor size in the hu-BLT mice in comparison to the NSG mice. Spleen-derived hu-BLT mouse NK cells were able to expand in the presence of autologous osteoclasts and substantially increase both cytotoxicity and secretion of IFN-γ, similar to those seen in peripheral blood-derived human NK cells, indicating that NK cells from hu-BLT mice are capable of expansion and functional activation when activating signals are given. Thus, the many similarities between human and hu-BLT mouse immune systems make this mouse model more appropriate to study human cancer. In particular, it is well-suited for studies of allogeneic NK cell-based immunotherapy in cancer treatment. The advantages and challenges of hu-BLT mice in cancer studies are also discussed in this report.

## 1. Introduction

The aim of this study is to demonstrate the different immune cell reconstitution and immune cells’ function in various tissues of NSG and humanized-BLT mice (hu-BLT). The procedure of human-immune engraftment to generate hu-BLT mice consists of the surgical implantation of human fetal liver pieces and thymus tissue under a renal capsule of NSG mice, followed by IV injection of autologous CD34^+^ hematopoietic cells [1,2,3]. In NSG-hu-BLT mice (hu-BLT mice developed from an NSG background strain), positive and negative selection during T cell development occurs in the human thymus, and immature T cells become functional CD4^+^ helper and CD8^+^ cytotoxic T cells after human MHC-class I and class II restrictions [4,5]. Further, in this mouse model, hematopoietic stem cells (HSCs) develop into human immune cells [6,7,8,9]. It was found that NSG-hu-BLT mice in comparison to NOD-*scid*-BLT mice exhibited substantially higher levels of human leukocyte reconstitution in their peripheral blood [10]. These characteristics prove that the NSG-hu-BLT model is the best available mouse model to study human immunity, and it also represents the most advanced and complete humanized mouse model generated to date [4].

We have previously shown a high reconstitution (40–80%) of human CD45^+^ immune cells in the peripheral blood, spleen, bone marrow, pancreas, and gingiva of hu-BLT mice [11,12,13]. Other studies have also shown human immune cells in the female reproductive tract, intestines, and rectum of hu-BLT mice [14,15]. An absence or impairment of NK cells and a lack of other immune cells in different immune-deficient strains could explain discrepancies in the ability of cancer stem cells (CSCs) to give rise to human tumors in those mice models [16]. This makes it hard to assess and compare the aggressiveness and metastatic potential of primitive CSCs in these mouse models [3]. We used hu-BLT mice to study the growth, metastasis, and therapies of human oral, melanoma, and pancreatic CSCs and established that this mouse model is the most suitable platform to implant such tumors [11,12] (manuscript in prep).

In the current study, we showed human and mouse CD45^+^ immune cells in the bone marrow, peripheral blood, and spleen of hu-BLT and NSG mice, respectively. Next, we implanted human oral and pancreatic tumors in NSG and hu-BLT mice and compared the disease progression and health status of mice over a period of four weeks. We also harvested tumors and compared tumor weights and tumor expansion in cultures for NSG and hu-BLT mice. Further, we compared the cytokine secretion and surface characteristics of tumors, as well as NK cell-mediated cytotoxicity against the tumors extracted from NSG and hu-BLT mice.

## 2. Materials and Methods

### 2.1. Tumor Cell Lines and Reagents

A tumor was surgically removed from oral cancer patients at UCLA, and oral squamous carcinoma stem cells (OSCSCs) were isolated from those tumors [17,18,19,20]. OSCSCs and oral tumors isolated from NSG and hu-BLT mice were cultured in RPMI 1640 (Life Technologies, Carlsbad, CA, USA) supplemented with 10% fetal bovine serum (FBS) (Gemini Bio-Product, West Sacramento, CA, USA). Dr. Nicholas Cacalano (UCLA David Geffen School of Medicine) kindly provided Mia PaCa-2 (MP2) human pancreatic cell lines. MP2 and pancreatic tumors isolated from NSG and hu-BLT mice were cultured in DMEM supplemented with 10% FBS and 2% penicillin-streptomycin (Gemini Bio-Products, West Sacramento, CA, USA). RPMI 1640 supplemented with 10% FBS was used to culture NSG and hu-BLT mice immune cells. Recombinant human IL-2 was purchased from Hoffman La Roche (Nutley, NJ, USA). Human IFN-γ ELISA kits were purchased from Biolegend (San Diego, CA, USA). Phosphate-buffered saline (PBS) and bovine serum albumin (BSA) were purchased from Life Technologies, Carlsbad, CA, USA. Matrigel was purchased from Corning, NY, USA.

### 2.2. Tumor Implantation in NSG and hu-BLT Mice

Animal research was conducted under the written approval of the UCLA Animal Research Committee (ARC) in accordance with all federal, state, and local guidelines. Combined immunodeficient NOD.CB17-Prkdcscid/J and NOD.Cg-Prkdcscid Il2rgtm1Wjl/SzJ (NSG lacking T, B, and NK cells) were purchased from Jackson Laboratory. Humanized-BLT (hu-BLT; human bone marrow/liver/thymus) mice were generated from NSG background as described previously [1,21]. For surgical implantation of tumors, hu-BLT and NSG mice were anesthetized with isoflurane in combination with oxygen. A total of 1 × 10^6^ human OSCSC and MP2 tumor cells suspended in 10 μL HC Matrigel were then injected directly into the floor of their mouths and in the pancreas, respectively. Mice were monitored for four to five weeks, and when signs of morbidity were evident, mice were euthanized. The tumors, bone marrow (BM), spleen, and peripheral blood were harvested.

### 2.3. Single-Cell Isolation of Tumors, Bone Marrow, Spleen, and Peripheral Blood of NSG and hu-BLT Mice

Single-cell suspensions were isolated from tumors, BM, spleen, and peripheral blood as described previously [22]. Briefly, oral and pancreatic tumors were immediately cut into 1 mm^3^ pieces and placed into a digestion buffer containing 1 mg/mL collagenase II (oral tumor) or collagenase IV (pancreatic tumor), 10 U/mL DNAse I, and 1% bovine serum albumin in DMEM and incubated for 20 min at 37 °C in an oven with a 150 rpm shaker. Next, the sample was filtered through a 70 µm cell strainer and centrifuged at 1500 rpm for 10 min at 4 °C. The pellet was re-suspended in DMEM media, and cells were counted. For BM, femurs were cut at both ends and flushed through using RPMI 1640 media; afterwards, cells were filtered through a 40 µm cell strainer. Spleens were minced, and the samples were filtered through a 40 µm cell strainer and centrifuged at 1500 rpm for 5 min at 4 °C. The pellet was re-suspended in ACK buffer for 2–5 min to remove the red blood cells followed by re-suspension in RPMI media and centrifugation at 1500 rpm for 5 min at 4 °C. For peripheral blood, PBMCs were isolated using Ficoll-Hypaque centrifugation of heparinized blood specimens. The buffy coats containing PBMCs were harvested, washed, and re-suspended in RPMI 1640 medium.

### 2.4. Human and hu-BLT NK Cells’ and Monocytes’ Purifications

All procedures were approved by the UCLA Institutional Review Board (IRB), and written informed consent approved by UCLA-IRB was obtained from healthy donors. Peripheral blood of healthy donors was used to isolate peripheral blood mononuclear cells (PBMCs) using Ficoll-hypaque centrifugation. Next, PBMCs were used to isolate NK cells and monocytes using the EasySep^®^ Human NK cell and EasySep^®^ Human Monocytes enrichment kits, respectively, purchased from Stem Cell Technologies (Vancouver, BC, Canada). For hu-BLT mice, splenocytes were used to isolate NK cells and BM cells were used to isolate monocytes using the EasySep^®^ Human NK cell and EasySep^®^ Human Monocytes enrichment kits, respectively, purchased from Stem Cell Technologies (Vancouver, BC, Canada).

### 2.5. Generation of Human and hu-BLT Osteoclasts

Monocytes were cultured in alpha-MEM media supplemented with M-CSF (25 ng/mL) and RANKL (25 ng/mL). The media were replenished every three days. Multinucleated cells (osteoclasts) on day 21 were used for experiments.

### 2.6. Expansion of Human and hu-BLT NK Cells

Human and hu-BLT NK cells were treated with rh-IL-2 (1000 U/mL) and anti-CD16 mAbs (3 µg/mL) overnight and were co-cultured with osteoclasts and sAJ2 probiotic bacteria (OCs:NK:sAJ2; 1:2:4) in RPMI 1640 medium containing 10% FBS. AJ2 is a combination of eight different strains of gram-positive probiotic bacteria (*Streptococcus thermophiles*, *Bifidobacterium longum*, *Bifidobacterium breve*, *Bifidobacterium infantis*, *Lactobacillus acidophilus*, *Lactobacillus plantarum*, *Lactobacillus casei*, and *Lactobacillus bulgaricus*). The media were refreshed every three days with RPMI complete medium containing rh-IL-2 (1500 U/mL).

### 2.7. NK Cells’ Supernatant Induced Differentiation of OSCSCs and MP2 Tumors

Human NK cells were treated with a combination of IL-2 (1000 U/mL) and anti-CD16 mAbs (3 μg/mL) overnight. The supernatant was harvested, and the levels of IFN-γ were assessed using single ELISA. Differentiation of OSCSCs and MP2 cells was conducted with an average total of 2000 to 3500 pg and 5000 to 7000 pg of IFN-γ from IFN-γ containing NK cells’ supernatants, respectively, over a period of 7 days.

### 2.8. Enzyme-Linked Immunosorbent Assays (ELISAs)

Single ELISAs were performed as previously described [23]. Briefly, in order to analyze and obtain IFN-γ concentration, a standard curve was generated by either two- or three-fold dilution of recombinant IFN-γ provided by the manufacturer.

### 2.9. 4-Hour Standard ^51^Cr Release Cytotoxicity Assay

A 4-h standard ^51^Cr release assay was performed as described previously [24]. Different concentrations of effector cells were incubated for 4 h with ^51^Cr–labeled tumor cells. Next, the supernatants were harvested from samples, and the released radioactivity was counted using the gamma counter. The percentage of specific cytotoxicity was calculated as follows:(1)$$\%\;\mathrm{Cytotoxicity}\;=\frac{\mathrm{Experimental}\;\mathrm{cpm}\;-\;\mathrm{spontaneous}\;\mathrm{cpm}}{\mathrm{Total}\;\mathrm{cpm}\;-\;\mathrm{spontaneous}\;\mathrm{cpm}}$$

Lytic units (LU) 30/10^6^ are calculated by using the inverse of the number of effector cells needed to lyse 30% of tumor target cells × 100.

### 2.10. Surface Marker Analysis Assay

For surface marker analysis, the cells were washed twice using ice-cold PBS + 1% BSA. For immune cell analysis, predetermined optimal concentrations of specific human or mouse monoclonal antibodies were added to 1 × 10^4^ cells in 50 µL of cold PBS + 1% BSA and were incubated on ice for 30 min. For tumors, predetermined concentrations of human MHC-class I and CD54 antibodies were added to 1 × 10^4^ tumor cells in 50 µL of cold PBS + 1% BSA and were incubated on ice for 30 min. Next, cells were washed in cold PBS + 1% BSA and brought to 500 µL with PBS + 1% BSA. Flow cytometric analysis was performed using a Beckman Coulter Epics XL cytometer (Brea, CA, USA), and results were analyzed in the FlowJo vX software (Ashland, OR, USA).

### 2.11. Statistical Analysis

GraphPad Prism-9 software was used for statistical analysis. An unpaired two-tailed student’s *t*-test was performed for the statistical analysis for experiments with two groups. One-way ANOVA with a Bonferroni post-test was used to compare different groups for experiments with more than two groups. (*n*) denotes the number of mice or human donors for each experimental condition. Duplicate or triplicate samples were used for cultures. **** (*p* value < 0.0001), *** (*p* value 0.0001–0.001), ** (*p* value 0.001–0.01), * (*p* value 0.01–0.05) symbols represent the levels of statistical significance within each analysis.

## 3. Results

### 3.1. Tumor-Bearing NSG Mice Lost Higher Body Weight in Comparison to Tumor-Bearing hu-BLT Mice

A similar number of tumor cells were injected into NSG and Hu-BLT mice in the pancreas or oral cavity, and disease progression was monitored for four weeks (Figure 1A). Human and mouse immune cells’ constitution was determined in bone marrow, peripheral blood, and the spleen of NSG and hu-BLT mice. NSG mice were found to have lower than 5% of mouse immune cells and no human immune cells, whereas no mice immune cells and greater than 58% human immune cells were seen in hu-BLT mouse tissues (Figure 1B,C and Appendix A). A significantly increased body weight loss was seen in oral tumor-bearing NSG mice in comparison to hu-BLT at week 2 and onwards (Figure 2A and Appendix A). In pancreatic tumor-bearing mice, a greater weight loss in NSG mice was seen on week 4 (Figure 2B and Appendix A). Since oral tumor-bearing mice were not able to eat properly in the presence of the tumor, their bodyweight loss was more significant as compared to pancreatic tumor-bearing mice (Figure 2 and Appendix A). To be able to study and compare these two groups, we euthanized mice at week four, because fast-growing tumors in tumor-bearing NSG mice resulted in mouse death four weeks after tumor implantation.

### 3.2. Higher Tumor Growth in NSG Mice in Comparison to hu-BLT Mice

Both NSG and hu-BLT mice were implanted with a similar number of tumor cells, as shown in Figure 1A. On week four, mice were euthanatized, tumors were resected, and tumor weight was determined. Tumor weight was significantly higher in both oral and pancreatic tumor-bearing NSG mice in comparison to hu-BLT mice (Figure 3A,B). We also observed higher melanoma tumor growth in NSG mice in comparison to hu-BLT mice (Appendix A).

### 3.3. Tumor Resected from NSG Mice Exhibited Higher Growth Rate in Cultures in Comparison to Those Resected from hu-BLT Mice

Tumors harvested from mice were dissociated and cultured overnight. All unattached/floating cells that contained the majority of immune effectors were removed, and the attached cells of any bound cells were washed with several rounds of PBS. After washing, we detached the attached cells with trypsin, and they were washed twice with media. With this process, we observed that only tumor cells were remaining, which we then counted, and an equal number of tumor cells was cultured for 14–20 days. Significantly higher tumor cell growth was observed in cultures derived from NSG mice in comparison to hu-BLT mice (Figure 4 and Appendix A).

### 3.4. Increased Level of IFN-γ Secretion Seen in Tumors Harvested from Hu-BLT Mice

We observed slight or no IFN-γ secretion in NSG mouse-derived tumors, whereas significantly higher IFN-γ secretion was seen in tumor cells derived from hu-BLT mice (Figure 5A,D). For CD45^+^ analysis, we stained the total populations of cells with anti-human CD45 antibody and gated on the population of lymphocytes using forward scatter (FS)/side scatter (SS) and determined the percentages of cells positive for CD45 expression within the gated population of lymphocytes. A significantly higher number of human CD45^+^ immune cells and IFN-γ secretion per one percentage of CD45^+^ immune cells was seen in hu-BLT mice in comparison to NSG mice (Figure 5B,C).

### 3.5. Tumors Isolated from NSG Mice Exhibited More Stem-like Phenotype in Comparison to Those from Hu-BLT Mice

Previous work from our laboratory has demonstrated that NK cell-mediated cytotoxicity is higher against stem-like tumor cells in comparison to their differentiated counterparts [17,18,23,25]. Here, we determined NK cell-mediated cytotoxicity against tumors derived from NSG and hu-BLT mice, and it was found that tumors derived from NSG mice were better lysed by NK cells compared to those from hu-BLT mice (Figure 6A,B). We have also previously shown that differentiated tumors exhibit increased CD54 and MHC-class I surface express levels [11,19,26]. Here, in this study, we found that tumors derived from hu-BLT mice expressed higher surface expression levels of MHC-class I (Appendix A). Results obtained for NK cell-mediated cytotoxicity and surface markers indicated that tumors isolated from hu-BLT mice were a little more differentiated in comparison to those obtained from NSG mice.

### 3.6. Tumors Expressing Higher MHC-Class I and CD54 on Their Surface Were Found Resistant to NK Cell-Mediated Cytotoxicity

We determined the cytotoxic function of NK cells against stem-like and differentiated oral and pancreatic tumors. The highest level of IFN-γ secretion was found when NK cells were treated with a combination of IL-2 and anti-CD16 mAbs (Figure 7A). As described in the Materials and Methods section, the supernatant of IL-2 + anti-CD16 mAbs-treated NK cells was used to induce differentiation in OSCSCs and MP2 tumors. NK-differentiated OSCSCs (Figure 7B) and MP2 (Figure 7E) tumors were found to be resistant to NK cell-mediated cytotoxicity and expressed a higher surface expression of MHC-class I (Figure 7C,F) and CD54 (Figure 7D,G).

### 3.7. hu-BLT NK Cells Are Capable of Expansion in the Presence of Osteoclasts Like Human NK Cells

We observed higher IFN-γ secretion and lower NK cell-mediated cytotoxicity in PBMCs of hu-BLT mice in comparison to human PBMCs when both PBMCs were treated with 1 million per 1 mL with 1000 U/mL IL-2 for 48 h (Table 1). To investigate the function of NK cells and osteoclasts in hu-BLT mice, we used autologous osteoclasts to expand NK cells (Table 2B). We used our established human NK expansion strategy to generate supercharged NK (sNK) cells [25] (Table 2A) and were able to expand hu-BLT spleen-derived NK cells (Table 2B). Significantly increased IFN-γ secretion and NK cell-mediated cytotoxicity were seen in both human and hu-BLT sNK cells as compared to their primary NK cell counterparts (Table 2 and Table 3). In hu-BLT mice, the order of IFN-γ secretion was seen as PBMCs > spleen > bone marrow > gingiva, but the order of NK cell-mediated cytotoxicity was seen as spleen > PBMCs > bone marrow > gingiva (Table 4).

## 4. Discussion

Mouse models of human diseases are crucial for our understanding of the mechanisms governing disease induction and progression, in particular for studies of cancer. The existing mouse models have some advantages but many important disadvantages. In this study, we compared oral and pancreatic tumor studies in NSG and hu-BLT mice. In addition, we compared the immune function of humans to hu-BLT mice to learn about the differences and similarities of the systems.

The NSG mouse model is the most frequently used mouse model for tumor studies mainly due to good engraftment of the tumors, ease of use and price point; however, as shown in this study, it is far from being the most appropriate model due to a lack of functioning immune cells, in particular, NK cells, monocytes, dendritic cells, and T and B cells. Immunodeficient mice such as the NSG do not recapitulate the complexity of the tumor microenvironment or the exact dynamics of tumor growth, since immune–tumor interaction shapes not only the growth dynamics of the tumor, but also its differentiation and survival, none of which takes place in NSG mice. To demonstrate the differences, we implanted oral and pancreatic tumors in NSG and hu-BLT mice and then compared the dynamics of tumor growth, rate of expansion, and differentiation among these two mouse models. Tumors grew much faster, and larger tumors formed in NSG mice when compared to hu-BLT mice. When tumors were dissociated, and single-cell tumor cells were cultured, the rate of expansion was much greater for tumors obtained from NSG mice than those from hu-BLT mice. Indeed, the rate of fold expansion for oral tumors ranged from 4.5–9.5 for NSG mice and 2.1–6.3 for hu-BLT mice when half a million tumor cells were cultured initially. The rate of tumor cell growth is greatly correlated with the increased MHC-class I expression on the tumor cells obtained from the hu-BLT (27–50% increase (Appendix A) and decreased NK cell-mediated cytotoxicity, a profile that we have previously shown to correlate with NK cell-mediated differentiation of the tumor cells in hu-BLT mice injected with allogeneic super-charged NK (sNK) cells [11,12]. Therefore, it is likely that immune cells in hu-BLT mice are not only suppressing the tumor growth by direct lysis of the stem-like tumors, but also by differentiating the tumor cells in vivo via secretion of IFN-γ and TNF-α, which limits the growth and expansion of tumor cells [19]. Indeed, when supernatants from NK cells are added to both oral and pancreatic stem-like tumors, the expression of differentiation antigens such as MHC-class I, CD54, and PDL-1 increases, the tumors grow much more slowly, and they become resistant to NK cell-mediated cytotoxicity [11,27,28]. This is directly relevant to induction, expansion, and tumor growth in vivo, since having competent immune cells is necessary for immune surveillance, lysis, and differentiation of the tumors. In the absence of any competent immune effectors, the tumors in NSG mice grow and expand, whereas in hu-BLT mice, there are enough competent cells to decrease the expansion and progression of the cancer. Indeed, in vivo NSG mice grow larger tumors, and the mice lose weight rapidly. Hu-BLT mice grow smaller tumors and lose weight at later times than NSG, but the tumors are still able to be established in them. This could be due to a number of reasons. Even though hu-BLT mouse peripheral blood has competent B and T cells, and the numbers and percentages are similar to humans [12], the percentages of NK cells are at approximately 50% those seen in human peripheral blood. Further, since both oral and pancreatic tumors are stem-like/poorly differentiated tumors, the percentages of NK cells are not enough to contain all the growing tumors. Indeed, when a million sNK cells were injected after tumor implantation, the tumor growth and expansion were substantially decreased [11,12]. In addition, the killing function of human NK cells in hu-BLT mice is approximately 50% less than those seen in the peripheral blood of humans, even though the levels of IFN-γ secretion are higher than those seen in the peripheral blood of humans (Table 1). Therefore, the decrease in both the percentages of NK cells and their killing function may facilitate the growth and expansion of stem-like/poorly differentiated aggressive tumors. There is considerable debate regarding the mechanisms of decrease in NK cells’ function in hu-BLT mice. It was hypothesized that deficiency in IL-15 may be one underlying mechanism for the decreased function and numbers of NK cells in hu-BLT mice [29,30]. Supplementation of IL-15 increased both the numbers and function of NK cells in hu-BLT mice [31]. Similar to hu-BLT, supplementation of IL-15 can also increases the cytotoxic function of NK cells and their numbers in human cancer patients [32,33]. In addition, we have not observed significant defects in the autologous NK cells’ function in hu-BLT mice, other than a reverse in the levels of cytotoxicity and IFN-γ secretion, which may point to the split anergy in NK cells in this mouse model system, since human NK cells may also have to deal with the mouse stromal cells. Split anergy denotes the concept of decreased cytotoxicity in the presence of increased IFN-γ secretion, which occurs after NK cells are cultured with tumor cells or after crosslinking of important receptors such as CD16 or NKp46 on NK cells [34]. Indeed, the CD56^bright^ population in the tissues, which most likely represents an activated phenotype of NK cells in humans, has also been shown to have decreased cytotoxicity in the presence of increased IFN-γ secretion [35,36]. Therefore, the conditioning of human NK cells in the context of the mouse tissue microenvironment may create the differences we observed between human-derived NK cells and hu-BLT-derived NK cells. To demonstrate the expansion and activation potential of NK cells from hu-BLT, we enriched splenocyte-derived NK cells and cultured them with the autologous osteoclasts differentiated from the monocytes. After different days of culture, we determined the expansion potential, as well as functions of NK cells, and found them to have excellent responsiveness, indicating that their activation and expansion potential is largely intact in the hu-BLT mice. The slight differences that were noted between the sNK cells from humans or hu-BLT mice is likely due to the compartments from which the NK cells were obtained: peripheral blood from the humans and spleen for hu-BLT. Due to a lack of an adequate volume of peripheral blood from hu-BLT, side-by-side comparisons of hu-BLT peripheral blood-derived NK cells with human peripheral blood-derived NK cells are challenging. However, these experiments are ongoing in our lab at the moment.

Even though we do see some differences related to the differentiation status of NK cells in humans and hu-BLT mice, the NK cells from hu-BLT mice are functional, and they were able to respond to signals from the stromal cells such as osteoclasts. Decreased percentages and cytotoxic function of NK cells in the presence of augmented IFN-γ secretion is likely important in the inhibition of tumor growth and decreased expansion in hu-BLT mice as compared to NSG mice, making this mouse model the best for studies of immunotherapy with the allogeneic NK cells, as we have shown previously in different tumor models [3,11,12]. Additional challenges exist when studying autologous T cell function in the hu-BLT mouse model, since implanted tumor cells are allogeneic and therefore might be important and relevant in studies of allogeneic T cell functions to implanted tumor cells, or it may necessitate the use of recipient HLA expression on the implanted tumor cells to study the autologous T cell function. All of these scenarios are under investigation in our laboratory at present.

## 5. Conclusions

Despite advances in therapeutic strategies, the five-year survival rate continues to remain low for the majority of cancers, suggesting an urgent need to find effective new mouse models to study the mechanisms of cancer progression as well as therapeutics. Mouse models of human disease are crucial for our understanding of the mechanisms governing disease induction and progression, particularly in studies of cancer. The existing mouse models have some advantages, but they also have many important disadvantages. In this study, we compared oral and pancreatic tumor studies in NSG and hu-BLT mice. In addition, we compared the immune function of humans to hu-BLT mice to learn about the differences and similarities of the systems.

The NSG mouse model is the most commonly used animal model for tumor studies, mainly due to good engraftment of the tumors, ease of use, and price point; however, we show in this study that it is far from being the most appropriate model due to a lack of functioning immune cells, in particular, NK cells, monocytes, dendritic cells, and T and B cells. Immunodeficient mice such as the NSG mice do not recapitulate the complexity of the tumor microenvironment or the exact dynamics of tumor growth, since immune–tumor interaction shapes not only the growth dynamics of the tumor, but also its differentiation and survival, none of which occurs in NSG mice.

Furthermore, the hu-BLT mouse model is one of the state-of-the-art technologies and tools in cancer research that has allowed us to evaluate the disease progression in relevant pre-clinical mouse models that closely resembles human disease. Not only do we have several published comprehensive reviews on the use of the hu-BLT mouse model in cancer, but we have also published many relevant manuscripts previously, despite the focus being on the use of hu-BLT mice alone and not a comparison between NSG and hu-BLT mice. Indeed, we are one of the few laboratories in the world that has access to hu-BLT mice. This paper demonstrates the differences between NSG and hu-BLT mice and points to the necessity for careful selection of mouse models to represent human disease if we want to obtain meaningful and interpretable data regarding human disease.

## Figures and Tables

**Figure 1 cancers-15-00112-f001:**
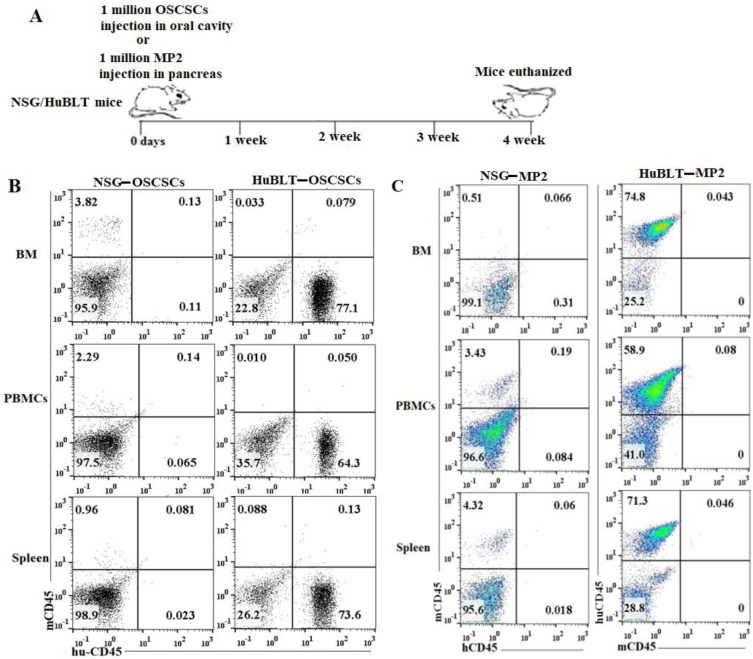
Experiment outline and immune cells’ reconstitution in NSG and hu-BLT mice. NSG and Hu-BLT mice were surgically implanted with 1 × 10^6^ tumor cells either in the pancreas or oral cavity, and disease progression was monitored for four weeks (**A**). After euthanasia, tissues were resected to obtain single cells. Percentages of human and mouse CD45^+^ immune cells were determined in single cells isolated from BM, PBMCs, and spleen. One of four representative experiments is shown in Figure 1 (**B**,**C**).

**Figure 2 cancers-15-00112-f002:**
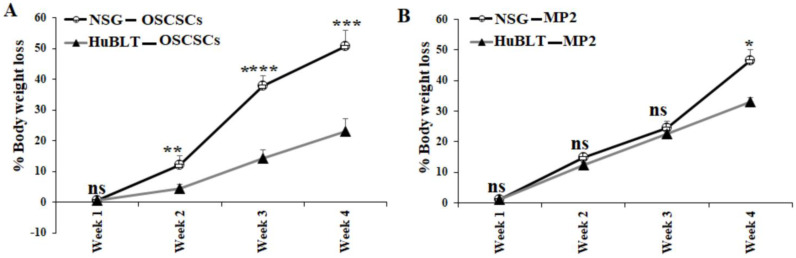
Higher bodyweight loss in NSG mice in comparison to hu-BLT mice after surgical implantation of OSCSCs or MP2 tumors. NSG and Hu-BLT mice were surgically implanted with 1 × 10^6^ tumor cells in the oral cavity ((**A**), *n* = 4) or pancreas ((**B**), *n* = 2), and the rate of percentages of body weight loss were determined on weeks 1, 2, 3, and 4. Symbols: **** (*p* value < 0.0001), *** (*p* value 0.0001–0.001), ** (*p* value 0.001–0.01), * (*p* value 0.01–0.05). ^ns^ no-significance represent the levels of statistical significance within each analysis.

**Figure 3 cancers-15-00112-f003:**
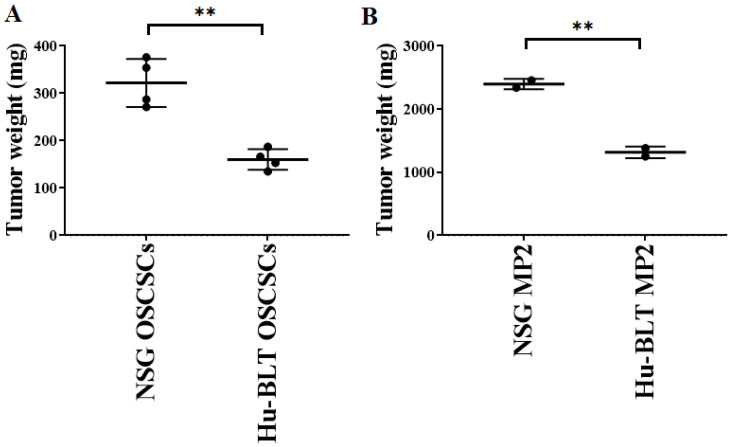
Significantly higher tumor growth in NSG mice in comparison to hu-BLT mice after surgical implantation of OSCSCs or MP2 tumors. NSG and Hu-BLT mice were surgically implanted with 1 × 10^6^ tumor cells in the oral cavity ((**A**), *n* = 4) or pancreas ((**B**), *n* = 2). On week 4, mice were euthanatized, tumors were resected, and tumor weight was determined. Symbols: ** (*p* value 0.001–0.01) represent the levels of statistical significance within each analysis.

**Figure 4 cancers-15-00112-f004:**
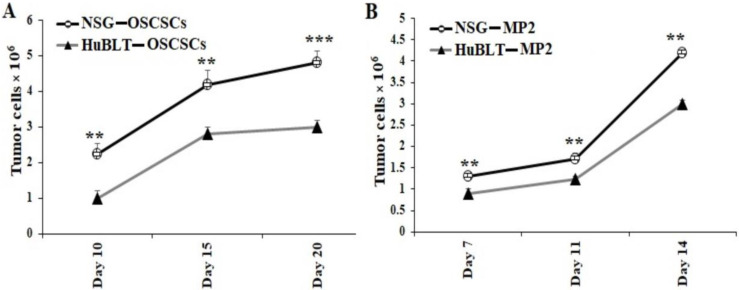
Tumors resected from NSG mice showed significantly higher tumor growth in cultures in comparison to those from hu-BLT mice. NSG and Hu-BLT mice were surgically implanted with 1 × 10^6^ tumor cells in the oral cavity ((**A**), *n* = 4) or pancreas ((**B**), *n* = 2). On week 4, mice were euthanatized, and tumors were resected and cultured (1 × 10^6^ cells/mL). Tumor cells were manually counted using a microscope on days as shown in the figures. Symbols: *** (*p* value 0.0001–0.001), ** (*p* value 0.001–0.01) represent the levels of statistical significance within each analysis.

**Figure 5 cancers-15-00112-f005:**
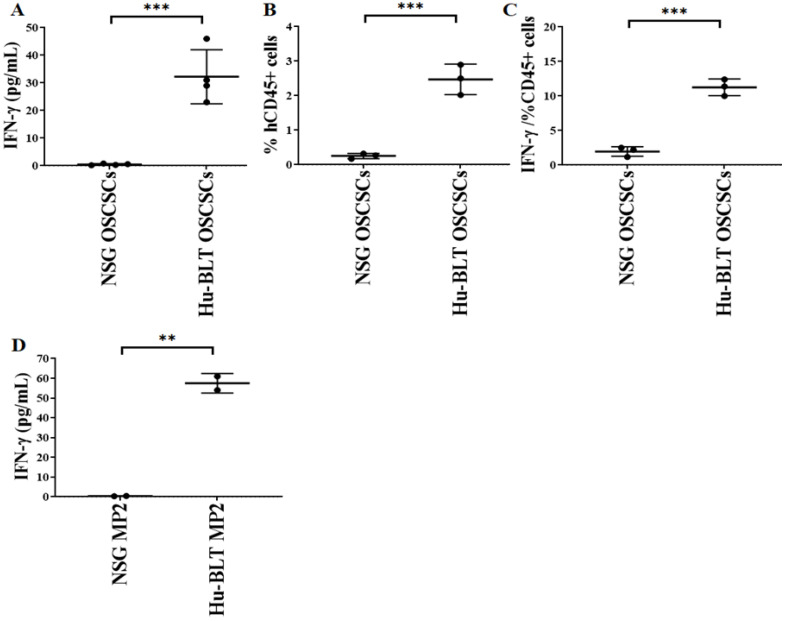
Significantly low IFN-γ was seen in tumor cultures of NSG mice in comparison to hu-BLT mice. NSG and Hu-BLT mice were surgically implanted with 1 × 10^6^ tumor cells in the oral cavity ((**A**–**C**), *n* = 3 or 4) or pancreas ((**D**), *n* = 2). On week 4, mice were euthanatized, tumors were resected, and similar numbers of tumor cells (1 × 10^6^ cells/mL) were cultured with IL-2 (1000 U/mL). Supernatants were harvested on day 7, and levels of IFN-γ were determined using single ELISA (**A**,**D**). Using a flow cytometer, the percentages of human CD45 within the mouse-derived oral tumors were determined (*n* = 3) (**B**). The percentages of human CD45 as shown in Figure B were used to determine the IFN-γ secreted by 1% of CD45^+^ immune cells in mouse-derived oral tumors (*n* = 3) (**C**). Symbols: *** (*p* value 0.0001–0.001), ** (*p* value 0.001–0.01) represent the levels of statistical significance within each analysis.

**Figure 6 cancers-15-00112-f006:**
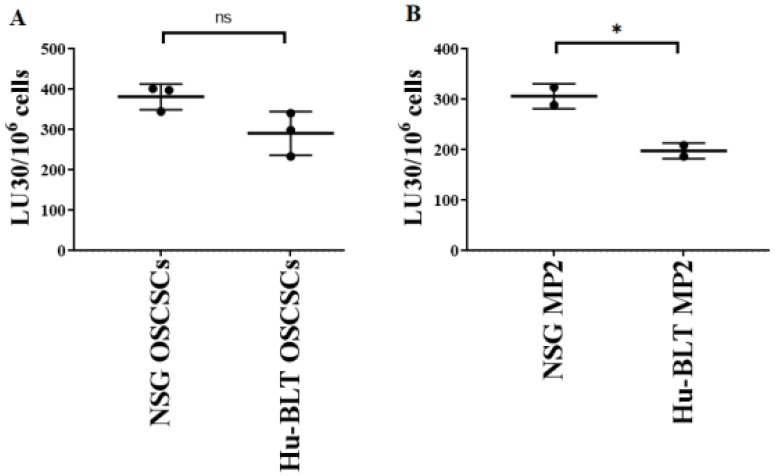
Tumors isolated from NSG mice exhibited a more stem-like phenotype in comparison to hu-BLT mice. NSG and Hu-BLT mice were implanted with 1 × 10^6^ tumor cells in the oral cavity ((**A**), *n* = 3) or pancreas ((**B**), *n* = 2). On week 4, mice were euthanatized, and tumors were resected and cultured (1 × 10^6^ cells/mL) for 7 days. Healthy human donors’ NK cells (1 × 10^6^ cells/mL) were treated with IL-2 (1000 U/mL) overnight and added to ^51^Cr-labeled tumors obtained from NSG or hu-BLT mice at different effector-to-target ratios for 4-h ^51^Cr release assay. The lytic units (LUs) 30/10^6^ cells were determined using the inverse number of NK cells required to lyse 30% of the tumor cells × 100. Symbols: * (*p* value 0.01–0.05), ^ns^ no-significance represent the levels of statistical significance within each analysis.

**Figure 7 cancers-15-00112-f007:**
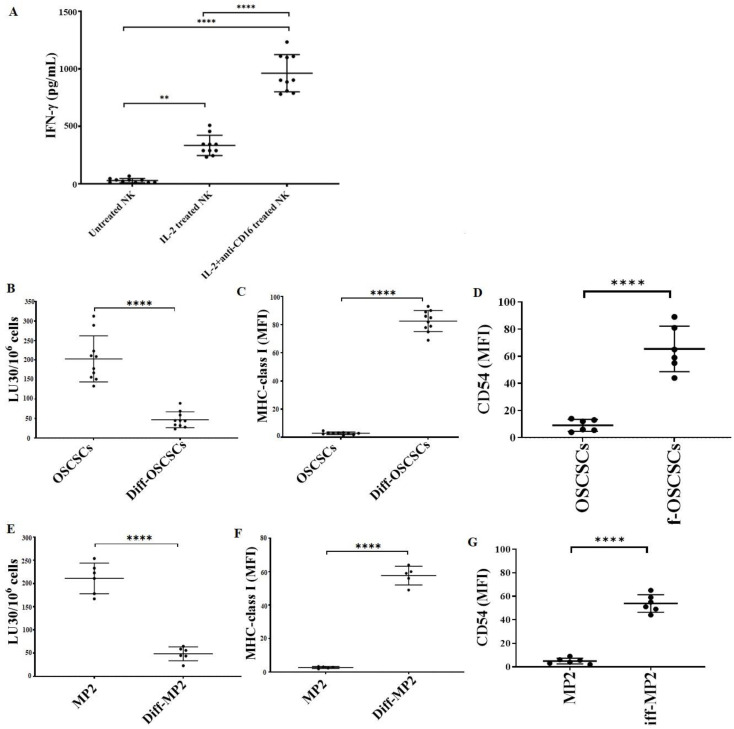
NK cell-mediated differentiated oral and pancreatic tumors expressed higher surface expression of MHC-class I and CD54 and were resistant to NK cell-mediated cytotoxicity. Healthy human donor-derived NK cells (1 × 10^6^ cells/mL) were left untreated, treated with IL-2 (1000 U/mL), or treated with a combination of IL-2 (1000 U/mL) and anti-CD16 mAbs (3 µg/mL) overnight. The supernatants were harvested, and the levels of IFN-γ secretion were determined using single ELISA (*n* = 10) (**A**). OSCSCs (**B**–**D**) and MP2 (**E**–**G**) tumors were differentiated using NK cells’ supernatants as described in Section 2.7. Healthy human donor-derived NK cells (1 × 10^6^ cells/mL) were treated with IL-2 (1000 U/mL) overnight and were added to ^51^Cr-labeled OSCSCs and Diff-OSCSCs (*n* = 10) (**B**) and MP2 and Diff-MP2 (*n* = 6) (**E**) at various effector-to-target ratios. The lytic units (LUs) 30/10^6^ cells were determined using the inverse number of NK cells required to lyse 30% of the tumor-cells × 100. Surface expression levels of MHC-class I (*n* = 10) (**C**) and CD54 (*n* = 6) (**D**) on OSCSCs, as well as MHC-class I (*n* = 5) (**F**) and CD54 (*n* = 6) (**G**) on MP2 tumors, were determined using flow cytometry. Symbols: **** (*p* value < 0.0001), ** (*p* value 0.001–0.01) represent the levels of statistical significance within each analysis.

**Table 1 cancers-15-00112-t001:** IFN-γ secretion and NK cell-mediated cytotoxicity by hu-BLT mice and human PBMCs. Freshly isolated PBMCs from peripheral blood of human and hu-BLT mice were treated with IL-2 (1000 U/mL) for 48 h before the supernatants were harvested and the levels of IFN-γ were determined using single ELISA. Freshly isolated PBMCs from peripheral blood of humans and hu-BLT mice were treated with IL-2 (1000 U/mL) for 48 h before they were added to ^51^Cr-labeled OSCSCs at different effector-to-target ratios. NK-mediated cytotoxicity was determined using 4-h ^51^Cr release assay. The lytic units (LUs) 30/10^6^ cells were determined using the inverse number of PBMCs required to lyse 30% of the tumor-cells × 100 (**A**). IFN-γ secretion and LUs 30/10^6^ cells per one NK cell % were determined using CD16% results of flow analysis (**B**).

A		
PBMCs	IFN-γ	LU 30/10^6^
**Hu-BLT (*n* = 6)**	1423 +/− 237	53 +/− 9
**Human (*n* = 6)**	546 +/− 79	185 +/− 81
**B**		
**PBMCs**	**IFN-γ/1% NK**	**LU 30/10^6^/1% NK**
**Hu-BLT (*n* = 6)**	189.7 +/− 142	7.1 +/− 4.3
**Human (*n* = 6)**	45.5 +/− 38	15.4 +/− 2.7

**Table 2 cancers-15-00112-t002:** Characteristics of human and hu-BLT mice NK cells. Freshly isolated NK cells (primary NK cells) of humans isolated from peripheral blood (**A**) and hu-BLT mice isolated from splenocytes (**B**) were treated with IL-2 (1000 U/mL) for 48 h before the supernatants were harvested and the levels of IFN-γ were determined using single ELISA. Freshly isolated NK cells (primary NK cells) of humans and hu-BLT mice were treated with IL-2 (1000 U/mL) for 48 h before they were added to ^51^Cr-labeled OSCSCs at different effector-to-target ratios. NK-mediated cytotoxicity was determined using 4-h ^51^Cr release assay. The lytic units (LUs) 30/10^6^ cells were determined using the inverse number of NK cells required to lyse 30% of the tumor-cells × 100. LUs 30/10^6^ cells per one NK cell % were determined using CD16% results of flow analysis. NK cells from humans (**A**) and hu-BLT (**B**) were treated with a combination of IL-2 (1000 U/mL) and anti-CD16 mAbs (3 µg/mL) overnight before they were cultured with autologous OCs in the presence of sAJ2 at a ratio of 1:2:4 (OCs:NK:sAJ2). On days shown in table, the numbers of cells were counted using microscopy, and fold expansion was determined using formula: number of cells recovered/number of cells cultured for each time point. The percentages of CD16^+^ and CD3^+^ cells in total cells were determined on days as shown in table. The supernatants were harvested from the co-cultures on days shown in table to determine the levels of IFN-γ secretion. Cytotoxicity on days shown in table was determined using a standard 4-h ^51^Cr release assay against OSCSCs. The lytic units (LUs) 30/10^6^ cells were determined using the inverse number of NK cells required to lyse 30% of the tumor-cells × 100. LUs 30/10^6^ cells per one NK cell % were determined using CD16% results of flow analysis.

A: Human (PBMC-Derived NK)	
	**Primary NK Cells**
IFN-γ secretion (pg/mL)	297 +/− 53
NK cell-mediated cytotoxicity (LU 30/10^6^ cells) LU 30/10^6^ cells/1% NK cell	228 +/− 72 2.49 +/− 1.7
	**Supercharged NK Cells**
Fold expansion (cells recovered/cells cultured)	Day 6: 1.4 +/− 0.6 Day 10: 3.1 +/− 0.3 Day 14: 3.8 +/− 1.1 Day 18: 4.6 +/− 1.2 Day 22: 3.01 +/− 0.7 Day 25: 1.94 +/− 0.2 Day 30:1.4 +/− 0.4 Day 34: 0.8 +/− 0.6
CD16 + cells %	Day 6: 93 +/− 3 Day 10: 90 +/− 7 Day 14: 89 +/− 4 Day 18: 92 +/− 2 Day 22: 91 +/− 6
CD3+ cells %	Day 6: 4 +/− 3 Day 10: 8 +/− 7 Day 14: 12 +/− 4 Day 18: 8 +/− 8 Day 22: 6 +/− 4
IFN-γ secretion (pg/mL)	Day 6: 3562 +/− 134 Day 10: 10,561 +/− 435 Day 14: 8972 +/− 276 Day 22: 4512 +/− 89
NK cell-mediated cytotoxicity (LU 30/10^6^ cells)	Day 10: 127 +/− 53 Day 14: 314 +/− 90 Day 18: 450 +/− 42
LU 30/10^6^ cells/1% NK cell	Day 10: 1.52 +/− 0.9 Day 14: 3.8 +/− 1.3 Day 18: 5.1 +/− 0.7
**B: Hu-BLT (Spleen-Derived NK)**	
	**Primary NK Cells**
IFN-γ secretion (pg/mL)	112 +/− 27
NK cell-mediated cytotoxicity (LU 30/10^6^ cells) LU 30/10^6^ cells/1% NK cell	113 +/− 31 2.28 +/− 0.56
	**Supercharged NK Cells**
Fold expansion (cells recovered/cells cultured)	Day 6: 2.2 +/− 0.1 Day 10: 3.4 +/− 1.2 Day 14: 2.8 +/− 0.7 Day 18: 1.9 +/− 0.3 Day 22: 0.78 +/− 0.2 Note: Hu-BLT NK cells expanded for day 22 only
CD16 + cells %	Day 6: 56 +/− 12 Day 10: 76 +/− 9 Day 14: 87 +/− 6 Day 18: 89 +/− 8 Day 22: 91 +/− 2
CD3+ cells %	Day 6: 47 +/− 11 Day 10: 23 +/− 17 Day 14: 21 +/− 13 Day 18: 14 +/− 9 Day 22: 5 +/− 4
IFN-γ secretion (pg/mL)	Day 6: 2344 +/− 38 Day 10: 3781 +/− 155 Day 14: 2678 +/− 209 Day 22: 1459 +/− 63
NK cell-mediated cytotoxicity (LU 30/10^6^ cells)	Day 10: 89 +/− 24 Day 14: 201 +/− 65 Day 18: 159 +/− 49
LU 30/10^6^ cells/1% NK cell	Day 10: 1.19 +/− 0.2 Day 14: 2.41 +/− 1.09 Day 18: 1.6 +/− 0.7

**Table 3 cancers-15-00112-t003:** Fold increase in IFN-γ secretion by supercharged NK cells vs. primary NK cells in humans and hu-BLT mice. Freshly isolated NK cells (Primary NK cells) of humans isolated from peripheral blood (**A**) and hu-BLT mice isolated from splenocytes (**B**) were treated with IL-2 (1000 U/mL) for 48 h before the supernatants were harvested and the levels of IFN-γ were determined using single ELISA. NK cells from humans (**A**) and hu-BLT (**B**) were treated with a combination of IL-2 (1000 U/mL) and anti-CD16mAb (3 µg/mL) overnight before they were cultured with autologous OCs in the presence of sAJ2 at a ratio of 1:2:4 (OCs:NK:sAJ2). On days shown in table, supernatants were harvested from the co-cultures to determine the amounts of IFN-γ secretion. Fold increase of IFN-γ secretion by supercharged NK cells vs. primary NK cells was determined for humans (**A**) and hu-BLT mice (**B**).

A	
sNK/Primary NK IFN-γ (*n* = 5)	Human (PBMC-Derived NK)
Day 6	10 +/− 3
Day 10	29 +/− 2.3
Day 14	25 +/− 1.7
Day 22	15 +/− 1.1
**B**	
**sNK/Primary NK IFN-γ (*n* = 5)**	**Hu-BLT (Spleen-Derived NK)**
Day 6	23 +/− 4
Day 10	32 +/− 2.5
Day 14	24 +/− 7
Day 22	15 +/− 2.1

**Table 4 cancers-15-00112-t004:** IFN-γ secretion and NK cell-mediated cytotoxicity by tissue compartments of hu-BLT mice. Freshly isolated PBMCs, bone marrow cells, splenocytes, and gingival cells of hu-BLT mice were treated with IL-2 (1000 U/mL) for 48 h before the supernatants were harvested and the levels of IFN-γ were determined using single ELISA. Freshly isolated PBMCs, bone marrow cells, splenocytes, and gingival cells of hu-BLT mice were treated with IL-2 (1000 U/mL) for 48 h before they were added to ^51^Cr labeled OSCSCs at various effector-to-target ratios. NK-mediated cytotoxicity was determined using 4-h ^51^Cr release assay. The lytic units (LUs) 30/10^6^ cells were determined using the inverse number of cells required to lyse 30% of the tumor-cells × 100.

Hu-BLT	IFN-γ	LU 30/10^6^
**BM (*n* = 6)**	43 +/− 6	28 +/− 6.3
**Spleen (*n* = 6)**	288 +/− 24	154 +/− 23
**PBMCs (*n* = 6)**	1423 +/− 237	53 +/− 9
**Gingiva (*n* = 2)**	12 +/− 4	9 +/− 4

## Data Availability

This study did not report any data.

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
