# Peer review of "Differences in Tumor Growth and Differentiation in NSG and Humanized-BLT Mice; Analysis of Human vs. Humanized-BLT-Derived NK Expansion and Functions"

_cancers, 2022, doi:10.3390/cancers15010112_

Round 1
Reviewer 1 Report
In this manuscript, the authors tested the differences between implanted tumor growth as well as differentiation in NSG and hu-BLT mice. Besides, the NK cells’ functions in hu-BTL mice were evaluated. Accordingly, the results indicated the appropriateness of hu-BLT mice model in human cancer studies.
Overall, the study is kind of interesting and has a set of data to support the conclusions. However, I have the following queries. The manuscript can be accepted after addressing the minor revisions noted.
1. Regarding Figure 1, there was kind of an obvious 0.9% population of cells that stained as double mCD45+hCD45+ in BM of Hu-BLT-MP2 mice. Please double check the data and explain it. Also, why there was no data of PBMCs in both NSG-MP2 and Hu-BLT-MP2? In addition, the authors described that “n = 4” in the Fig caption. Thus, please provide other related Flow cytometry profiles. Here it is highly suggested that the authors could show all the data with error bar in a separate figure which will be much easier for the readers to capture and understand. Also, please supplement the Flow gating strategies.
2. In terms of the experiments on the evaluation of the growth rate for the resected tumor cells from both NSG and Hu-BLT mice, the authors described that “tumor harvested from mice were dissociated and similar numbers of tumor cells were cultured for 14-20 days”. Here since the tumor tissues resected from Hu-BLT mice should have decent number of the tumor infiltrated immune cells (e.g., human T cells and NK cells), how to make sure the numbers of tumor cells (1X106 cells/ml) were same between NSG and Hu-BLT groups due to the interference of these infiltrated immune cells? Whether the authors sorted tumor cells out, then counted? Anyway, please make it clear to readers.
3. In Figure.5, the authors pointed out that “Using flow cytometer, the percentages of human CD45 within the mice-derived oral tumors were determined (n=4)”. Please provide the Flow gating strategies. Besides, only two repeats (n =2) were shown in Figure.5D, and the authors showed a statistical analysis on it. Please explain how?
4. The authors pointed out that they have shown differentiated tumors exhibit increased CD54 and MHC-class I surface express levels. While here only the MHC-I expression were indicated. Please add the CD54 expression if possible. Also, no related detailed methods were described. For example, how to measure the MHC-I expression, through Flow cytometry?
5. The title of the manuscript is kind of two long. Please refine it. Also, since there is no NK cells in NSG mice, thus it is not appropriate to say “NK cell function in NSG mice”.
6. The entire manuscript might need to be carefully edited to enhance the clarity and conciseness and to eliminate grammatical and syntax errors.
Author Response
We are very thankful to reviewer and here is point by point respond to all comments:
- Regarding Figure 1, there was kind of an obvious 0.9% population of cells that stained as double mCD45+hCD45+ in BM of Hu-BLT-MP2 mice. Please double check the data and explain it. Also, why there was no data of PBMCs in both NSG-MP2 and Hu-BLT-MP2? In addition, the authors described that “n = 4” in the Fig caption. Thus, please provide other related Flow cytometry profiles. Here it is highly suggested that the authors could show all the data with error bar in a separate figure which will be much easier for the readers to capture and understand. Also, please supplement the Flow gating strategies.
Response: We thank the reviewer for the comment. The 0.9% is non-specific binding. To avoid any issues with interpretation we have now replaced the figure with another experiment in which we demonstrate PBMCs for both NSG-MP2 and BLT-MP2. We have also added scatter graph error bars using four experiments in Supplementary file (Fig. S1).
- In terms of the experiments on the evaluation of the growth rate for the resected tumor cells from both NSG and Hu-BLT mice, the authors described that “tumor harvested from mice were dissociated and similar numbers of tumor cells were cultured for 14-20 days”. Here since the tumor tissues resected from Hu-BLT mice should have decent number of the tumor infiltrated immune cells (e.g., human T cells and NK cells), how to make sure the numbers of tumor cells (1X106 cells/ml) were same between NSG and Hu-BLT groups due to the interference of these infiltrated immune cells? Whether the authors sorted tumor cells out, then counted? Anyway, please make it clear to readers.
Response: We cultured the tumor samples overnight, removed all unattached/floating cells which contain the majority of immune effectors, washed attached cells of any bound cells with several rounds with PBS. After washing we detached the attached cells with trypsin and washed twice with media. With this process we observed that only tumor cells were remaining, which we then counted, and equal number of tumor cells were cultured. This information is now added in result 3.3.
- In Figure.5, the authors pointed out that “Using flow cytometer, the percentages of human CD45 within the mice-derived oral tumors were determined (n=4)”. Please provide the Flow gating strategies. Besides, only two repeats (n =2) were shown in Figure.5D, and the authors showed a statistical analysis on it. Please explain how?
Response: We stained the total populations of the cells with anti-human CD45 antibody and for the analysis we gated on the population of lymphocytes using FS/SS and determined the percentages of cells positive for CD45 expression within the gated population of lymphocytes. All the other regions in FS/SS did not demonstrate any CD45+ population, therefore, we knew based on FS/SS we were gating on the right population of CD45+ cells. This information is now added result 3.4. For these graphs we performed unpaired T test.
- The authors pointed out that they have shown differentiated tumors exhibit increased CD54 and MHC-class I surface express levels. While here only the MHC-I expression were indicated. Please add the CD54 expression if possible. Also, no related detailed methods were described. For example, how to measure the MHC-I expression, through Flow cytometry?
Response: We have now added CD54 both for oral and pancreatic tumors in Fig. 7, and also described methodology in section 2.10.
- The title of the manuscript is kind of two long. Please refine it. Also, since there is no NK cells in NSG mice, thus it is not appropriate to say “NK cell function in NSG mice”.
Response: We revised the title as suggested.
- The entire manuscript might need to be carefully edited to enhance the clarity and conciseness and to eliminate grammatical and syntax errors.
Response: We carefully edited and corrected errors in the entire manuscript.

Reviewer 2 Report
The manuscript by Kaur and Jewett presents the differences in tumor growth between NSG and hu-BLT mice. Also, the authors assess the functionality of hu BLT NK cells vs human NK. The manuscript is well-written and easy to follow.
Please see below my comments:
Do the tumors recovered from NSG vs hu BLT show differences in their histology?
The authors establish the advantages of hu BLT mice, in terms of their immune reconstitution, as models for cancer research, Could the authors show the immunoprofiling of the tumors recovered from the hu BLT mice? Do these tumors have immune infiltration? Other than NK, what other cell types could be acting directly to slow tumor growth?
Figures 2 and 4. To reduce visual overload, I consider that the bar graphs could be grouped into a line graph to show the full time-course, eg. one for OSCsC and one for MP2 tumors.
Author Response
We are very thankful to the reviewer, and here is point by point responses to comments:
Do the tumors recovered from NSG vs hu BLT show differences in their histology?
Response: We have conducted histology on some of the BLT tumors but not on all tumors. However, we have done flow analysis of the tumors and shown differences not only on immune infiltration but also the differentiation markers of CD44, CD54, MHC class I and PDL-1. This information is already published (Kaur et al. Probiotic-Treated Super-Charged NK Cells Efficiently Clear Poorly Differentiated Pancreatic Tumors in Hu-BLT Mice. Cancers. 2020; 12(1):63. https://doi.org/10.3390/cancers12010063, and Kaur et al. Super-charged NK cells inhibit growth and progression of stem-like/poorly differentiated oral tumors in vivo in humanized BLT mice; effect on tumor differentiation and response to chemotherapeutic drugs, OncoImmunology, 7:5, DOI: 10.1080/2162402X.2018.1426518 ).
The authors establish the advantages of hu BLT mice, in terms of their immune reconstitution, as models for cancer research, Could the authors show the immunoprofiling of the tumors recovered from the hu BLT mice?
Response: This has been published previously (Kaur et al. Probiotic-Treated Super-Charged NK Cells Efficiently Clear Poorly Differentiated Pancreatic Tumors in Hu-BLT Mice. Cancers. 2020; 12(1):63. https://doi.org/10.3390/cancers12010063, and Kaur et al. Super-charged NK cells inhibit growth and progression of stem-like/poorly differentiated oral tumors in vivo in humanized BLT mice; effect on tumor differentiation and response to chemotherapeutic drugs, OncoImmunology, 7:5, DOI: 10.1080/2162402X.2018.1426518)
Do these tumors have immune infiltration? Other than NK, what other cell types could be acting directly to slow tumor growth?
Response: Yes, T cells infiltrate the tumors and these have been shown in our previous publications (Kaur et al. Probiotic-Treated Super-Charged NK Cells Efficiently Clear Poorly Differentiated Pancreatic Tumors in Hu-BLT Mice. Cancers. 2020; 12(1):63. https://doi.org/10.3390/cancers12010063, and Kaur et al. Super-charged NK cells inhibit growth and progression of stem-like/poorly differentiated oral tumors in vivo in humanized BLT mice; effect on tumor differentiation and response to chemotherapeutic drugs, OncoImmunology, 7:5, DOI: 10.1080/2162402X.2018.1426518 ). We know that supercharged NK cells expand CD8+ T cells and this concept is very important for the infiltration of both NK and CD8+ T cells in the tumor microenvironment. We have published in our previous paper that without sNK cells the percentages of NK cells within the tumor microenvironment is significantly less than when we inject sNK cells. In contrast, the percentages of T cells are much higher in non sNK injected tumors and in sNK injected tumors the percentages of T cells is decreased (Kaur et al. Osteoclast-expanded super-charged NK-cells preferentially select and expand CD8+ T cells. Sci Rep 10, 20363 (2020). https://doi.org/10.1038/s41598-020-76702-1). Therefore, both cells are crucial for tumor growth control, although they will target different clones of the tumor cells, NK cells will target CSC/poorly differentiated and differentiate tumor cells and CD8+ T cells will target the differentiated tumors that express higher levels of MHC-class I.
Figures 2 and 4. To reduce visual overload, I consider that the bar graphs could be grouped into a line graph to show the full time-course, eg. one for OSCsC and one for MP2 tumors.
Response: We have now added line graphs for Fig. 2 and 4, and moved previous version to supplementary file.
